# *De-novo* protein function prediction using DNA binding and RNA binding proteins as a test case

Sapir Peled[1], Olga Leiderman[1], Rotem Charar[1], Gilat Efroni[1], Yaron Shav-Tal[1] & Yanay Ofran[1]

Of the currently identified protein sequences, 99.6% have never been observed in the laboratory as proteins and their molecular function has not been established experimentally. Predicting the function of such proteins relies mostly on annotated homologs. However, this has resulted in some erroneous annotations, and many proteins have no annotated homologs. Here we propose a *de-novo* function prediction approach based on identifying biophysical features that underlie function. Using our approach, we discover DNA and RNA binding proteins that cannot be identified based on homology and validate these predictions experimentally. For example, FGF14, which belongs to a family of secreted growth factors was predicted to bind DNA. We verify this experimentally and also show that FGF14 is localized to the nucleus. Mutating the predicted binding site on FGF14 abrogated DNA binding. These results demonstrate the feasibility of automated *de-novo* function prediction based on identifying function-related biophysical features.

[1] The Goodman Faculty of Life Sciences, Nanotechnology building, Bar Ilan University, Ramat Gan 52900, Israel. Correspondence and requests for materials should be addressed to Y.O. (email: Yanay@ofranlab.org).

Many studies attempt to make sense of the tremendous amounts of new genomic sequences by analysing DNA sequences. However, since biological processes are executed predominantly by proteins, to decipher biological function one needs to go beyond genomic sequences and analyse the proteins these sequences encode. Unfortunately, the rate of sequencing is not matched by the rate of annotation of the function of proteins[1]. Experimental annotation of the molecular function of proteins typically requires expression and purification of the protein. This is difficult to perform on a large-scale, and often fails for many proteins. Currently, 99.6% of the entries in UniProtKB[2] describe proteins that were never observed experimentally as a protein. Some of them were observed only as RNA transcripts and others are hypothetical proteins or predicted from DNA sequence. Computational protein function prediction is thus one of the only avenues for narrowing the ever-growing gap between sequence data and biological insight[3]. An assessment of existing methods for automated annotation of protein function has concluded that there is considerable need for improvement of currently available tools[4].

About 40% of the functional annotations of proteins in the Gene Ontology (GO)[5,6] are predicted based on homology, using annotation transfer. To predict the function of a newly discovered protein, this approach searches for a homologous protein whose function is known, assuming that the similarity in sequence reflects also similarity in function. But large-scale assessments of this approach disprove this assumption[7,8]. It has been shown, for example, that even for sequences with extremely high sequence similarity (BLAST E-values $<10^{-70}$), homology based annotation predicts a wrong function 60% of the time[8]. Moreover, many proteins do not have known homologs, and others have only unannotated ones. Therefore, de-novo prediction methods, which do not rely on homology to annotated sequences, would often be the only route. Unfortunately, there is currently no systematic way to predict molecular function de-novo. Some function prediction tools attempt to infer function from the 3D structure of the protein[9]. However, experimentally solved structures are available for only 0.07% of all UniProtKB entries. Therefore, a sequence-based prediction may be the only solution for most known proteins.

The challenge of function prediction has interesting similarities and differences to the challenges of protein structure prediction. De-novo structure prediction was proven to be very hard[10], requiring extensive resources[11] and reaching only limited success. Homology based structure prediction, on the other hand, has high success rates and is fairly easy to implement. Even low levels of sequences similarity enable good prediction of protein structure[10]. For function prediction, however, homology based predictions yield dubious results[7,8]. Can protein function be predicted de novo from sequence? It has been suggested[1] that this may be possible by focusing on functional sites (e.g., binding sites, catalytic sites). Many methods were designed to identify functional sites, provided that the function of the protein is already known. They can, therefore, provide additional functional insight into an already annotated protein, but cannot annotate an un-annotated one. We hypothesized that since functional sites define the molecular function of the protein and are composed of residues that possess specific biophysical characteristics, it may be possible to use them as a basis for an automated de-novo function prediction.

Nucleic-acid (NA) binding proteins (NABPs) constitute a useful test case for such approach. They are involved in vital cellular processes (transcription, recombination, replication, DNA packing, modification and repair) and are defined by their ability to bind single- or double-stranded DNA or RNA. Thus, DNA or RNA binding sites, if successfully identified, may be an emblem that gives away the function of a protein.

It is believed that many NA binding proteins in the genome have not been discovered yet[12]. Numerous computational methods were developed to predict whether a given protein binds NA. Some rely on overall sequence or structure homology to other NABPs, while others look for homology through shorter sequence signatures or similarity in traits such as amino acid composition[4,13,14]. Other methods do not try to predict whether a protein binds NA. Rather they focus on proteins that are already known to bind NA and attempt to predict which residues make up the binding site. Over the last fifteen years, dozens of computational methods of these two types have been introduced. Due to their number it is impossible to review all of them here but many of them are discussed in recent reviews[15,16]. The success of these methods notwithstanding, they are limited in their scope and applicability: The first class of methods, namely methods for the discovery of proteins that bind NA, is usually limited to those proteins that have overall similarity to known NABPs. The second, namely methods that identify putative NA binding sites, is typically relevant only to proteins that are already known to bind NA and the methods can help identify the binding site within them. Both approaches are not optimized for de-novo function prediction, i.e., the discovery of novel NABPs, and the respective binding sites, among proteins that are not homologues of known ones.

Methods for high-throughput identification of protein-NA interactions, such as ChIP-seq[17] and RIP-seq[18], produce substantial amounts of data but their large scale applicability is hampered by costs and the requirement of prior knowledge about the proteins (e.g., availability of antibodies). They are also prone to errors[19,20]. We developed two methods, one for identifying novel DNA binding proteins (DBPs) and the other for identifying RNA binding proteins (RBPs), even if they have no homology to known ones. Both methods also predict the NA binding residues. Our approach acts in two stages: given a query sequence, a set of machine learning models first searches for residues that possess characteristics that are typical of DNA binding residues. While such residues may occur also in proteins that do not bind DNA, we found that their distribution is different in DNA binding proteins than in non-DNA binding ones. Thus, a second set of machine learning models predicts whether, given the distributions of predicted DNA binding residues, the query protein itself binds DNA. A similar set of machine-learning-based models was designed to identify proteins that bind RNA. In the task of identifying novel NA binding proteins this approach for de-novo prediction succeeds where most existing tools fail. We demonstrate this success by identifying proteins that are not yet known to bind DNA and showing experimentally that they do bind DNA. The method also identifies the binding site and we demonstrate experimentally that mutating the predicted residues in this binding site hampers, or abrogates, NA binding. We call this method Dr PIP: DNA or RNA Protein Interaction Prediction. It is available at: www.ofranlab.org, under services.

## Results

**Prediction of DNA- and RNA-binding residues.** One RF model was trained to predict DNA binding residues, and another RF model was trained to predict RNA binding residues. Figure 1a shows the precision-recall curve (PRC) for predicting DNA-binding residues. To train these machines, each residue was described by 246 features that included: predicted structural features (secondary structure, solvent accessibility and disorder) as well as sequence features (type of amino acid, sequence neighbors and their characteristics, conservation, entropy. See Methods for details). Figure 1b presents the PRC for predicting RNA-binding residues. The light-grey line represents the expected performance of a random guess. The PRC shows a tradeoff

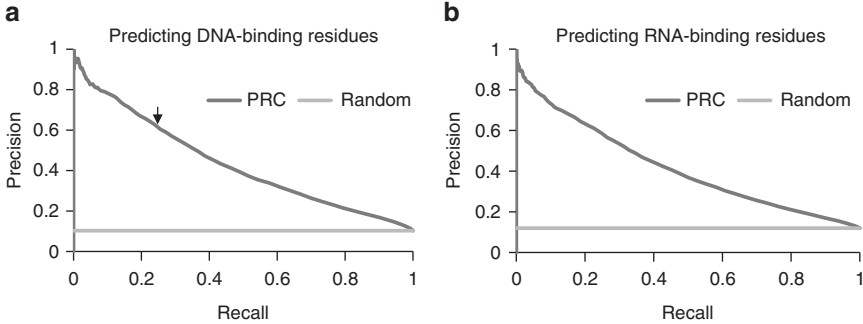

**Figure 1 | Precision-recall curves (PRC) for predicting NA binding residues.** (**a**) PRC for DNA-binding residues prediction. Arrow marks the cutoff score that yielded precision of 0.65 and recall of 0.23 (**b**). PRC for RNA-binding residues prediction.

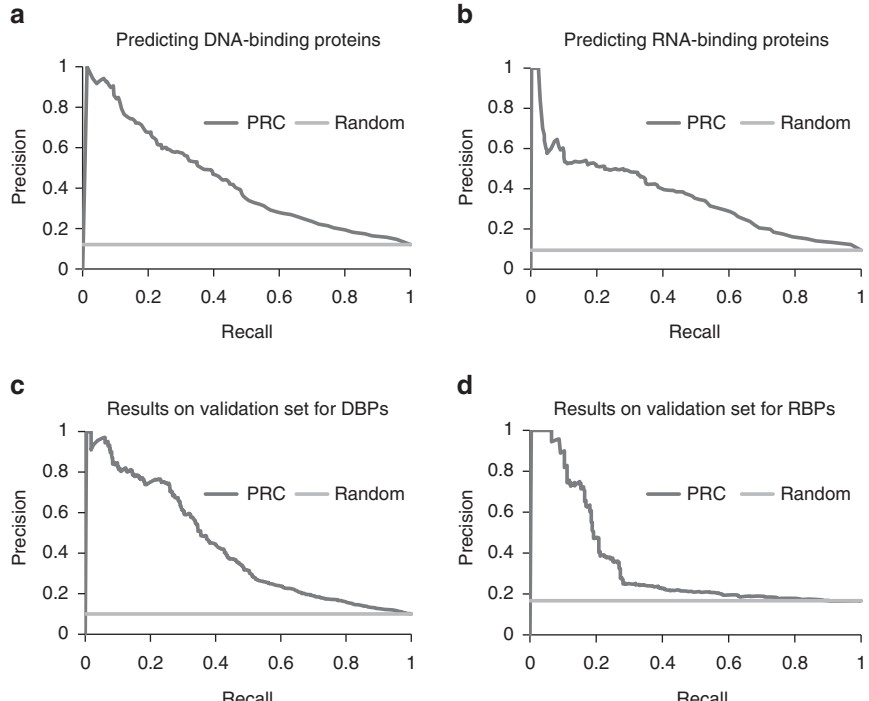

**Figure 2 | Precision-recall curves (PRC) for predicting NA binding proteins.** We assessed the PRC of each prediction method on two datasets: cross-validation on the set that was used to train the method, and an independent validation set (**a**) predicting whether a protein binds DNA, using cross-validation with the original set. (**b**) predicting whether a protein binds RNA, using cross-validation with the original set. (**c**) predicting whether a protein binds DNA, using an independent validation set. (**d**) predicting whether a protein binds RNA, using an independent validation set.

between the measures; high precision at low recalls and vice versa. For example, in Fig. 1a an arrow points to the performance of a cutoff score that yielded precision of 0.65 (i.e., at this cutoff ∼65% of the residues that are predicted to bind DNA are indeed observed experimentally to bind DNA), recall was 0.23 (that is, at this cutoff, ∼23% of the residues that are observed experimentally to bind DNA were identified as such by the model). One can choose any point along this curve as the operation point for the prediction method.

**Prediction of DNA- and RNA-binding proteins**. The per-residue predictions described above classify each residue as a potential DNA binding site, potential RNA binding site, or neither. However, residues that appear as putative NA binding sites may sporadically occur also in proteins that do not actually bind DNA. To predict whether the protein as a whole can bind DNA or RNA we trained and tested two additional RF models based on the prediction generated by the first two classifiers. To

train these machines, each protein was described by 15 features representing the distribution and the scores of the prediction of the per-residue machines (e.g., how many of the residues got a certain score. See Methods). The assumption behind these models is that the distribution of predicted binding residues can differentiate proteins that actually bind NA from proteins that do not.

Figure 2a shows the precision-recall curve (PRC) for DNA-binding proteins prediction and Fig. 2b presents the PRC for the RNA-binding proteins prediction. ROC curves are shown in Supplementary Figs 1 and 2. Area under curve (AUC) for DBPs prediction in 0.77 and for RBPs prediction is 0.79. Detailed assessment of their performance is provided in Supplementary Tables 1 and 2.

Importantly, the DNA binding prediction model was also tested against the positive RBPs dataset, and was found to give lower scores to RBPs and higher scores the DBPs (Supplementary Fig. 3). The RNA binding prediction model gave higher scores to RBPs than to DBPs (Supplementary Fig. 4).

**Dr PIP performed well for the validation sets**. These two prediction models were pipelined to create a prediction method that takes as input a sequence, predicts binding residues and uses these predictions to predict whether or not the protein binds DNA. A similar pipeline was created for predicting RNA binding sites and RNA binding proteins. We call this method DNA / RNA Protein Interaction Predictor (Dr PIP). The method was trained on NABPs whose structures were solved experimentally. To validate its performance in identifying NABPs even when the structure is not known, we used validation sets of DBPs and non-DBPs that have no solved 3D structure and were not used for training or testing. Similar sets were used for testing and validating the models for predicting RBPs and non-RBPs (Method). Figure 2c,d show the performance of Dr PIP on these independent validation sets.

**Dr PIP distinguished between DBPs and non-DBPs**. For further validation we set to analyse some of the predicted DBPs experimentally. The experimental validation we seek must be available even when we don't know which RNA or DNA sequence the protein binds. For protein-DNA interactions, B1H[21] system can provide such validation. Thus, we focused on validating predicted DBPs. We selected proteins for which we could not find positive or negative experimental information regarding their interaction with DNA. The set of proteins we assembled for this experimental analysis is presented in Table 1. None of these proteins had detectable sequence similarity to any protein in the positive set. However, FGF18 was found to be similar to FGF19 that is part of the negative set. For each of these proteins we predicted whether they are expected to bind DNA. We used B1H for validating these predictions. For positive

control we used a known DNA binding protein, Zif268 (ref. 22) (Supplementary Table 9 and Supplementary Fig. 5) and then we applied B1H to determine experimentally whether the predicted proteins bind DNA. The results of both prediction and experiments are presented in Table 1. The seven proteins that received the highest scores were also found to be DNA binding. Interestingly, although sequence-similar to a protein in the negative set, FGF18 received a very high score and indeed was observed experimentally to bind DNA, showing that Dr PIP can identify DBPs in cases where homology based prediction would have failed.

**Additional validation using FGF14**. Fibroblast growth factors make up a large family of growth factors that are found in organisms ranging from nematodes to humans. In vertebrates, there are 22 members of the FGF family. Most FGFs have a signal peptide and are known to be secreted[23,24]. Our previous analysis suggested and verified that FGF18 binds DNA. We wanted to further explore members of the family. As opposed to most FGFs, FGF14 does not have a signal peptide for secretion. We applied Dr PIP to FGF14 and it was predicted as to have 52.9% probability to be DNA binding.

Subsequently, FGF14 was tested experimentally for DNA-binding using the B1H system. Figure 3 shows the prediction score per residue for FGF14. The residue with the highest score is residues 186, marked with an arrow. The three residues N-terminal to it created a stretch of strongly predicted residue, suggesting that they are a crucial part of the DNA binding site. We tested how mutating these residues affects DNA binding. In a B1H experiment an interaction between the tested protein and DNA leads to bacterial growth on minimal medium that lacks histidine and contains 3-amino-triazole (3-AT), which is a competitive inhibitor of HIS3. Increasing concentrations of 3-AT represent increasingly stringent binding conditions and hence are expected to yield reduced number of colonies. As shown in the second line (B) of Table 2, the WT FGF14 bounds DNA. This is reflected in the large number of colonies on the plate and in the fact that the number of colonies grows as the concentration of the 3-AT decreases. Site-directed mutagenesis in the predicted binding residues abrogated FGF14 DNA binding almost entirely.

**FGF14 is present in the nucleus**. The sub-cellular localization of FGF14 protein was examined in human U2OS cells after transfection with GFP-FGF14. As seen in Fig. 4, the FGF14 wildtype and FGF14 mutant proteins localized specifically in the nucleus compared to the control GFP protein that was diffusely distributed in all cell compartments.

**Table 1 | Experimental B1H[21] results for DNA binding and Dr PIP prediction of the experimental data set.**

| Protein | B1H result | Dr PIP Score |
|---------|-----------|--------------|
| FGF18 | Binding | 0.929 |
| GSCR2 | Binding | 0.842 |
| CS043 | Binding | 0.794 |
| IFRD1 | Binding | 0.766 |
| CI009 | Binding | 0.747 |
| CC130 | Binding | 0.476 |
| CHOO4 | Binding | 0.45 |
| Mak16 | Non- binding | 0.255 |
| CN093 | Non- binding | 0.184 |
| SOCS4 | Non- binding | 0.133 |

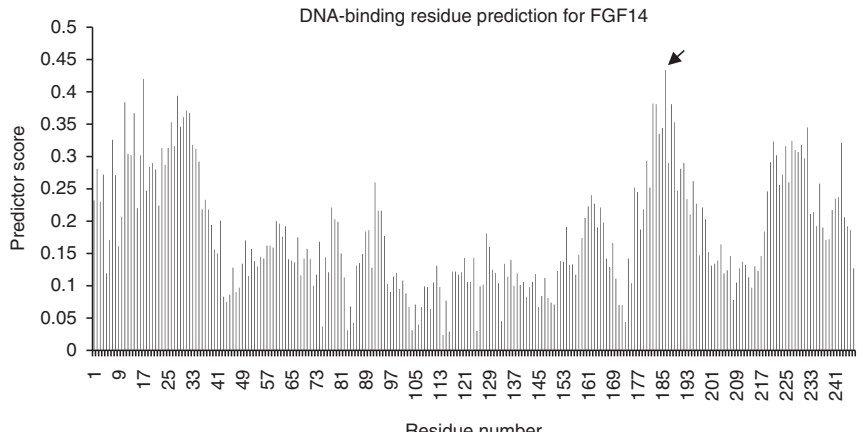

**Figure 3 | Score per residue from the DNA binding residue prediction for FGF14.** Arrow marks the residue that got the highest score.

**Table 2 | Number of surviving colonies on selective 3-AT plates using B1H assay.**

| | | Efficiency | 3-AT [mM] | | |
|---|---|---|---|---|---|
| | | | 2 | 3 | 4 |
| a | pB1H1 + pH3U3 lib | $10^8$ | 0 | 0 | 0 |
| b | pB1H1-FGF14 + pH3U3 lib | $10^8$ | 1,200 | 750 | 550 |
| c | pB1H1-FGF14(186–191) + pH3U3 lib | $10^8$ | 8 | 3 | 0 |

Survival represents DNA binding. (**a**) A plasmid without FGF14 (pB1H1) does not bind DNA. (**b**) A plasmid with FGF14 shows DNA binding activity. (**c**) FGF14 in which four residues that were predicted by Dr PIP to be in the binding site were mutated, yielded a negligible number of colonies suggesting virtual abrogation of binding.

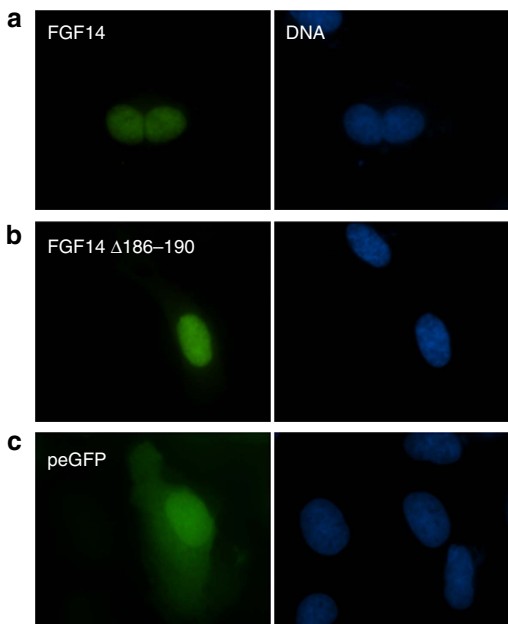

**Figure 4 | Localization of the GFP-FGF14 fusion proteins in transiently transfected U2OS cells.** (**a**) wildtype (**b**) mutant (**c**) control GFP. Green- fusion proteins, blue- Hoechst stain. Scale bar, 20 μm.

**Comparing Dr PIP to other methods in *de-novo* prediction.** The advantage of the approach proposed here is its potential for *de-novo* prediction. To demonstrate this, we constructed a set of proteins that were shown experimentally to bind DNA but have no homologs that are known to bind DNA. These proteins can test the *de-novo* predictions as other methods that rely on homology will not be able to identify them as DBPs. We identified ORFan proteins (i.e., proteins with no known homologues) whose function was studied experimentally. Some of them were shown to bind DNA and others were not. We then submitted each of these proteins to Dr PIP and to other publically available tools for predicting DNA binding proteins[25–30]. The results of this comparison are shown in Fig. 5 (detailed list of proteins and prediction scores is in Supplementary Table 4). Dr PIP was the only method that comprehensively distinguished between DNA binding and non-binding proteins. That is, all the DBPs got higher scores that any of non-DBPs.

**Comparison to other methods.** We used a previously published benchmark[29] to compare Dr PIP to existing servers. Table 3 shows the performance of several methods in this benchmark compared to that of Dr PIP.

To allow for a more detailed comparison of precision-recall curves, we focused on methods that give numerical scores rather than binary classification. We submitted our own set of DBPs to DNABIND[26] and DNAbinder[27]. The PRC is shown in Fig. 6a. Dr

PIP performance on the non-redundant dataset significantly exceeds the other methods.

**Comparing to other methods on novel DBPs.** Dr PIP was designed to excel on proteins that have no similarities to known NABPs. Thus, we created a set of proteins that are known to bind DNA, but are not members of DNA-related superfamilies, nor have any DNA-related sequence motif or profile (see Methods). The set contained 331 proteins. For the negative dataset we used the previously described DBPs negative validation set. Figure 6b shows that performance of Dr PIP compared to the other methods.

## Discussion

Only 11% of UniProtKB entries have a function description in their comment section, leaving the vast majority of the proteins with no functional annotation. Even proteins that have functional annotation may have additional, yet unknown, function[31,32]. Hence, many protein functions are waiting to be uncovered. Protein function is a general term for at least three different ontological aspects of biology, as defined by GO: biological process, molecular function and cellular location. Our approach is relevant for molecular function prediction. A significant fraction of the proteome is made up by DBPs. In the human proteome, estimates of the percentage of transcription factors (TFs) alone range from 6 to 12% (ref. 12) of all gene products. Currently in UniProtKB, for every TF there are two other DBPs that are not TFs. Given this proportion, the percentage of all kinds of DNA binding proteins may be about 18–36% of all proteins. However, currently less than 1% of experimentally annotated gene products were found to bind DNA. The gap between the meager 1% of known proteins that are verified experimentally as DNA binding and the expected fraction of up to 36% is currently filled by predicted DBPs based on homology. However, large scale analysis has shown that even at extremely high similarity levels (e.g., with BLAST E-value of $10^{-120}$) most of the homology based annotations are erroneous[8]. Moreover, since so few of the estimated DBPs have been discovered, there may be undiscovered families of DBPs. CAFA, the large scale assessment of automated methods for function prediction[4], defined a target as 'difficult' if it had sequence identity of 60% or less to an annotated protein. However, even more difficult are proteins with no detectable identity to any annotated protein. *De-novo* prediction may reveal them. The FGF family, for example, is not expected to bind DNA. It is named after members of the family that are extracellular factors. The stringent filters we applied delineated members of the family as negative examples in the training set. Dr PIP, however, identified two members of the family as DNA binding, a prediction that was corroborated experimentally both by B1H and by observing FGF14 in the nucleus, showing that DNA binding is possible under biological conditions. Indeed, a high throughput analysis of putative DBPs, listed some members of the family as positive examples[33].

While the same computational approach is applicable to DBPs and RBPs, availability of data and experimental tools allowed us

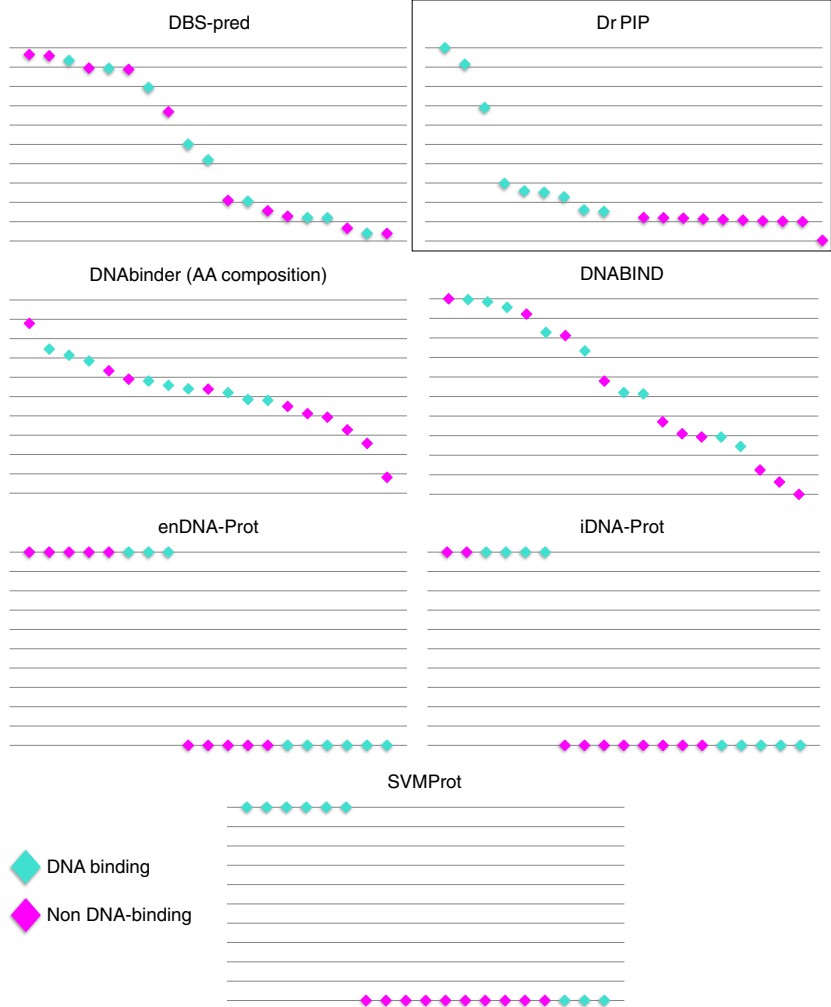

**Figure 5 | Predictions of different methods for a set of proteins with no homology to known DNA binders.** Cyan diamonds represent proteins annotated as DNA-binding that have no homology to any other known DBPs. Magenta diamonds represent proteins from the negative set of proteins that are unlikely to bind DNA. On the Y-axis is the score of the prediction method. Proteins are ordered by their score. Some methods provide binary predictions. Dr PIP fully separates positive and negative samples, with all binders with the higher scores and all the non-binders with lower scores.

**Table 3 | Performance of top performing methods on an independent dataset[29].**

| Method | ACC (%) | MCC | s.e. (%) | SP (%) | F1-M (%) |
|---|---|---|---|---|---|
| DNAbinder (P21) | 79.00 | 0.61 | 54.87 | 98.08 | 70.31 |
| DNAbinder (P400) | 80.11 | 0.62 | 58.53 | 97.97 | 72.73 |
| DNA-Prot | 84.61 | 0.69 | 73.17 | 94.00 | 81.08 |
| iDNA-Prot | 77.47 | 0.55 | 78.05 | 77.00 | 75.73 |
| enDNA-Prot | 84.62 | 0.70 | 73.18 | 94.00 | 84.62 |
| **Dr PIP** | **96.10** | **0.92** | **92.60** | **99.00** | **95.55** |

Dr PIP performance on the same dataset was also added, for comparison.

to validate DNA binding proteins more extensively. B1H offers a quick experimental corroboration for DBP predictions without requiring knowledge of the DNA binding site and without requiring purification of the protein. We could not find a similar method for assessing RBP predictions. As shown in Fig. 2a,b, the performance of the DBP prediction models was better than that of the RBPs prediction models. A possible explanation is that RNA can fold into intricate secondary structures, increasing the structural diversity of the interfaces. DNA has less structural variation (not considering, of course, chromosomal organization that is less relevant for the size of interfaces we consider here); hence the DNA binding interfaces may be more similar to one another and easier for the machine algorithm to learn.

Looking at the amino acid composition of residues in the interface can also provide some insight into the biophysical characteristics that allow NA recognition. Supplementary Table 8 shows the fraction of positive, negative, polar and hydrophobic residues, as well as that of residues that can form hydrogen bonds. It shows these fractions in proteins in general, in the interfaces and in residues predicted by Dr PIP to bind DNA or RNA. The composition of the predicted interfaces is virtually indistinguishable from that of observed interfaces, indicating that the machine learned the correct composition of interfaces. Not surprisingly, positive residues are strongly overrepresented while negative residues are under-represented, indicating that the most pronounced feature of the interfaces is the electrostatic interaction.

We used one experimental method to validate Dr PIP as a prediction tool. Specific proteins, of course, may require custom tailored experiments to provide conclusive evidence regarding their functions.

We see the main strength of Dr PIP in its ability to predict function *de-novo*. That is, predicting NABPs that could not be

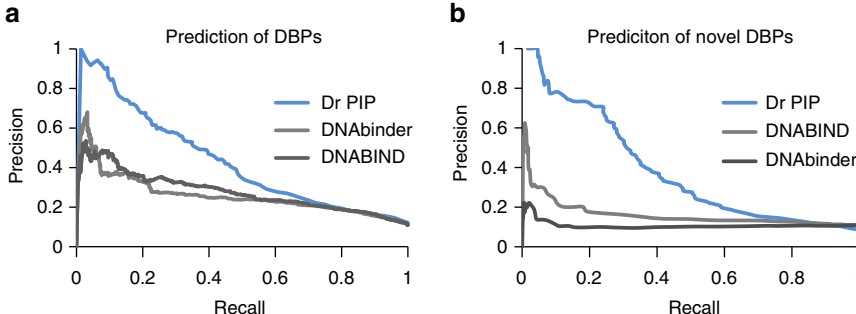

**Figure 6 | Precision-recall curves (PRC) of the performances of Dr PIP compared to two publically available DNA binding protein prediction methods:** DNAbinder[21] and DNABIND[20], (**a**) on the non-redundant dataset of DBPs and non-DBPs that was used for Dr PIP construction, (**b**) on a set of novel DBPs, which are not members of DNA-binding superfamilies and do not have DNA binding motifs, patterns or profiles.

predicted based on similarities to known NABPs. However, we first assessed its performance on a previously published bench-mark dataset[29]. As shown in Table 3, Dr PIP outperformed existing methods. For two of these methods we also constructed detailed PRCs on our own dataset. We then performed specific tests to assess the performance of Dr PIP on novel DBPs. On a dataset of ORFans, namely proteins that have no known homologs at all, Dr PIP was the only methods to fully separate between DNA-binding ORFans and non-DNA binding ORFans. To provide an assessment on a larger scale, we used a dataset of proteins that are not ORFans, but are not members of known DNA binding family and have no sequence motif or profile that is associated with DNA binding. We compared Dr PIP to two other non-binary prediction methods on this dataset and showed that while other methods performance of this set was only slightly above random, Dr PIP's performance was close to its performance on the other sets.

Of the many existing computational methods for the analysis of NABPs, some predict functional residues, others predict protein function for the whole protein, and a few provide both predictions. We show that it is possible not just to provide both predictions, but also to rely on the former to predict the latter, supporting out hypothesis that protein molecular function is defined by its functional sites. While we demonstrate the feasibility of this approach to predicting RNA and DNA binding proteins, sequence-based *de-novo* function prediction can be further implemented to the prediction of other protein functions, and provide a large-scale annotation of proteins, so long that the function is characterized by a functional site.

## Methods

**Creation of NA-binding protein datasets.** Structures that contain both protein and DNA chains were extracted from the RCSB PDB website (http://www.rcsb.org/)[34] filtering by molecule type on the advanced search options. Fasta-format sequences of the proteins were obtained from the SEQRES lines. Using the coordinates in the ATOM line, all protein-NA contacts (<5Å between NA and protein atoms, hydrogen excluded) were mapped. Protein chains with no such contacts were removed. Sequences shorter than 30 residues and sequences with undetermined residues, labeled as 'X', were removed. Sequences were then clustered using BLASTclust[35] (30% identity, 50% length coverage) to form a non-redundant dataset. The same process was repeated for structures that contain both protein and RNA. This resulted in a set of 513 DNA binding protein chains and a set of 389 RNA-binding protein chains (proteins are listed at Supplementary Tables 5 and 6).

**Residue feature vector construction.** Each residue was described as a set of calculated and predicted numeric attributes. They include: Evolutionary con-servation extracted per each residue and the number of proteins in the alignment, extracted from HSSP[36]; predicted solvent accessibility and secondary structure from PROF-phd[37]; predicted disorder from MD[38]. In addition, since a residue is affected by its neighbors, sliding windows were used as descriptors of each residue. The properties of the surrounding residues were also included in the vector. We explored different sizes of sliding windows for each attribute to determine the optimal window. For the secondary structure and disorder, a window size of seven

residues, centered by the residue for which we wish to provide prediction, gave the best performance. For the evolutionary profile, a window size of nine residues was used. Each vector contained a total of 246 features. Each such vector described one residue and was labeled: '1' or '0' for a binding or a non-binding residue, respectively.

**Filtration of residue feature vectors.** Protein sequences were aligned to the sequence from the PDB structure using EMBOSS Needle global alignment tool[39]. Residues that are not present in the crystal structure were removed, since they cannot be labeled as binding or non-binding. These residues were still included as descriptors of other residues (in sliding windows) if they are not the central ones.

**Training and testing of binding site prediction model.** Vectors were divided into 3 non-redundant groups for 3-fold cross validation. The machine learning algo-rithm that was chosen is random forest (RF)[40], which yielded better results on our dataset than neural networks (NN) and support vector machine (SVM). Training was done with 1,000 trees.

**Prediction results analysis.** Prediction scores for the test set were compared to the observed classifications, where a score (between 0 and 1) was considered as positive (i.e., binding) if it was above a certain cutoff. Precision and recall, as defined below, were calculated for different cutoff values, and a precision-recall curve (PRC) was created for each model.

$$\text{Precision} = \frac{TP}{TP + FP}$$

$$\text{Recall} = \frac{TP}{TP + FN}$$

Where TP is the number of true positive predictions, FP is the number of false positives and FN is the number of false negatives. Expected performance at random was calculated as the fraction of positive examples from the total number of examples (which usually is equal to the precision value when recall = 1). Sensitivity (true positive rate), specificity, false positive rate (FPR), accuracy (ACC) and Matthews correlation coefficient (MCC) as defined below, were calculated for different cutoff values. A receiver operating characteristic (ROC) curve was created for each model using TPR and FPR values. Expected performance at random is a diagonal line from the left bottom corner to the top right one. Area under curve (AUC) was calculated to farther evaluate performances.

$$\text{Sensitivity(TPR)} = \frac{TP}{TP + FN}$$

$$\text{Specificity} = \frac{TN}{TN + FP}$$

$$\text{FPR} = \frac{FP}{TN + FP}$$

$$\text{ACC} = \frac{TP + TN}{TP + FP + TN + FN}$$

$$\text{MCC} = \frac{TP \times TN - FP \times FN}{\sqrt{(TP + FP)(TP + FN)(TN + FP)(TN + FN)}}$$

Where TN is the number of true negative predictions.

**NABPs positive datasets.** To train the secondary prediction model, the one that predicts whether a whole protein binds NA we used the same positive datasets described above.

**Table 4 | Genes whose ORFs were cloned into the pB1H1 vector and their restriction sites.**

| Gene | pB1H1 restriction site |
|------|------------------------|
| *FGF18_HUMAN* | *NotI/AvrII* |
| *GSCR2_HUMAN* | *KpnI/BamHI* |
| *CS043_HUMAN* | *KpnI/AvrII* |
| *IFRD1_HUMAN* | *KpnI/BamHI* |
| *CI009_HUMAN* | *NotI/AvrII* |
| *CC130_HUMAN* | *NotI/AvrII* |
| *CH004_HUMAN* | *KpnI/BamHI* |
| *MAK16_HUMAN* | *KpnI/BamHI* |
| *CN093_HUMAN* | *KpnI/BamHI* |
| *SOCS4_HUMAN* | *NotI/AvrII* |

**Negative datasets creation.** All SwissProt proteins were downloaded from the UniProtKB website (release 2015_05). Sequences shorter than 30 residues were removed, along with sequences containing undetermined residues. Remaining SwissProt entries were scanned for terms related to nucleic acids and cellular processes they are involved in, e.g., transcription, and for GO terms 'DNA binding' and 'RNA binding'. Entries containing at least one of these terms were removed. To match the positive set, all proteins with no solved structure were removed (the validation set below has proteins with unsolved structure). Remaining sequences were clustered using BLASTclust with same parameters mentioned above for the positive dataset, resulting in 4151 non-binding protein chains. This filter does not guarantee that this set does not include NABPs, however it is likely to be heavily enriched by non-NABPs.

**Protein feature vector construction.** Based on the DNA- and RNA- binding residues predictions, a set of features was composed per protein. They included the following attributes:

1. Length of the protein
2. Highest prediction score
3. Number of resides with prediction score above 0.7
4. Number of resides with prediction score above 0.6
5. Number of resides with prediction score above 0.5
6. Number of resides with prediction score above 0.4
7. Number of resides with prediction score above 0.3
8. Number of resides with prediction score above 0.2
9. Number of resides with prediction score above 0.1
10. Number of sequence windows of 4 residues in which at least two of the residues received a prediction score above 0.3.
11. Number of residues with predictions above/bellow a Z-score of 3.
12. Number of residues above a Z-score of 2.
13. Number of residues above a Z-score of 1.
14. Number of residues above a Z-score of 0.5.
15. Number of residues bellow a Z-score of $-2$.

The same was done in the training of RNA-binding prediction model.

**Training and testing of binding protein prediction model.** Training was performed using RF with 1,000 trees and 3-fold cross validation. Performance was assessed as above.

**DNA validation set.** *Positive dataset.* All proteins from SwissProt containing 'DNA binding' GO term (go:0003677) were taken and filtered for direct experimental evidence codes in order to obtain a dataset of experimentally verified DNA binding proteins. Redundancy was checked within the dataset itself and against the training set using the parameters that were used in the dataset construction, as mentioned above, and removed. Thus, no sequence in the validation set was redundant with any other sequence in the validation set or with any other sequence in the training set. This resulted in 557 protein chains (listed in Supplementary Table 7).

*Negative dataset.* Negative set was created the same as the one for the DNA- and RNA- binding proteins prediction model, however, only proteins with no solved structure were taken. Redundancy against the negative set was checked and removed. A representative set of 5,000 non redundant proteins was then chosen.

**RNA validation set.** *Positive dataset.* As described in the DNA validation set, all proteins from SwissProt containing 'RNA binding' GO term (go:0003723) were taken and filtered for direct experimental evidence codes in order to obtain a dataset of experimentally verified DNA binding proteins. Redundancy was removed as for the DNA. This resulted in 318 protein chains.

*Negative dataset.* The negative set used for the DNA validation set was also used here.

**Defining a combined score using validation set prediction.** The cross validation yielded three models. To provide a single prediction per sequence we used a majority function, i.e., a protein was deemed positive at the lowest precision threshold for which at least two out of the three models would characterize it as positive. This score will be the final output of Dr PIP.

**ORFans set.** The HSSP[36] database of was downloaded, and proteins with only one protein in the alignment (that is, themselves) were selected. For the positive set, only proteins containing NA-binding relevant terms as previously described were taken. This resulted in a set of nine proteins. For the negative set, from all proteins that did not contain any NA-binding relevant terms, ten proteins were used.

**Novel DBPs.** We extracted novel DBPs from the positive DNA validation set before redundancy reduction. Each protein containing the word 'DNA' in its UniProtKB documentation of superfamilies, patterns and profiles (provided by the InterPro[41], PANTHER[42], Pfam[43], PRINTS[44], SMART[45] and PROSITE[46] databases) was removed. Of the original set of 1,032 known DBPs we were left with 608 proteins. After redundancy reduction with 30% similarity cutoff and 90% coverage for the training set of Dr PIP we were left with 331 proteins.

The negative DBPs dataset describes in 4.3.1 was used.

**Experimental analysis set.** For experimental validation of the predictions of Dr PIP, we ran Dr PIP on human proteins that are not known to bind DNA. From those we searched ones that were present in a library of human cDNA provided to us courtesy of Dr Doron Gerber (BIU). A set of ten proteins that represent the range of prediction scores was chosen as representatives of the larger data set for the experimental analysis.

**Reagents and instrumentation.** The chemicals used in Bacterial one-hybrid (B1H) studies were of molecular biology grade and were purchased from Sigma (MO, USA). Restriction enzymes were purchased from NEB. Chemically synthesized DNA oligonucleotides were ordered from IDT DNA. Components of the B1H system, reporter vector pH3U3-mcs, expression vector pB1H1 and the US0ΔhisBΔpyrF *E. coli* selection strain, were obtained from Professor Scot Wolfe through Addgene (http://www.addgene.org).

**Bacterial one hybrid system.** The binding site library was designed with a single nucleotide 5′ ACTGCGGCCGCGTCTTCAAACGCGTGTACACCTATCAG(N) 18GACTATGGCGCGCCATACTACTA. The library was amplified with a set of primers For-5′ ACTGCGGCCGCGTCTTCAAA and Rev-5′ TAGTAGTATGG CGCGCCATA. The reporter pH3U3 plasmid library containing randomized 18 bp binding sites was prepared as described in refs 21 and 47. The bait plasmid, pB1H1 was prepared as follows: ORF of genes listed in Table 4 were amplified by PCR from a human cDNA library (a generous gift from Doron Gerber) and cloned into the pB1H1 vector between appropriate restrictions sites (primers are listed in Supplementary Table 3).

**Bacterial one hybrid validation.** Prior to the selection, system was validated using known DNA binding protein Zif268 as described in ref. 21. US0ΔhisBΔpyrF *E. Coli* competent cells containing one of the Zif268 plasmid were transformed with the either empty PH3U3 plasmid or pH3U3 plasmid containing the library. The cells were spread on plates containing NM medium[47] with 2, 3 and 4mM 3-AT and incubated for several days in 37 °C until the colonies became visible. 15 colonies were randomly peaked from the 4 mM 3-AT plate and pH3U3 − 18N region was sequenced. Motif logo was generated using MEME algorithm[48].

Negative control with empty pB1H1 plasmids combined with either empty or library containing pH3U3 plasmid was used in each B1H experiment.

**Bacterial one hybrid protein analysis.** For binding site selection, US0ΔhisBΔ pyrF *E. Coli* competent cells containing one of the bait plasmids were transformed with the pH3U3 library. The cells were spread on plates containing NM medium[47] with 1, 2 and 4mM 3-AT and incubated for several days in 37 °C until the colonies became visible.

**FGF14 mutant cloning.** FGF RxKKT186-190A × AAA mutant was constructed using Site Directed Mutagenesis Kit (Stratagene) according to the manufacturer's protocol using FW primer 5′ GCT ATG AAA GGG AAC AGC GTA GCG GCA GCC AAA CCA GCA GCT C 3′ and RV primer 5′ GAG CTG CTG GTT TGG CTG CCG CTA CGC TGT TCC CTT TCA TAG C 3′. Mutated FGF14 was ligated into pB1H1 bait vector between KpnI and BamHI sites.

**FGF14 localization.** In order to assay the localization of FGF14 in the cell, we cloned the ORF of FGF14 wildtype and mutant Δ186–190 into the pEGFP-c1 vector between KpnI and BamHI restriction sites. The resulting fusion proteins were transfected into human U2OS osteosarcoma cells (ATCC) using the Lipofectamine 2000 (Invitrogen). The expression of the proteins was examined 24 h

after transfection and after fixation in 4% formaldehyde (20 min). The nucleus was co-stained with Hoechst. Images were acquired on an Olympus IX-81 wide-field fluorescent microscope at 60× magnification.

**Data availability.** The authors declare that the data supporting the findings of this study are available within the article and its supplementary information files.

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

## Acknowledgements

We thank Noa Kinor (BIU) for assistance with the experiments and Doron Gerber (BIU) for primers and genes. We also thank Shauli Ashkenazi and Anat Burkovitz (former ofranlab members) for help with the server, Sharon Fishman (Biolojic Design Ltd.) for commenting on the manuscript and Guy Nimrod (Biolojic Design Ltd.) for commenting on the server.

## Author contributions

Y.O. conceived the study. Y.O., S.P., O.L. and Y.S.-T. designed the experiments. S.P., O.L., R.C. and G.E. performed the experiments. Y.O. and S.P. analysed the results and wrote the paper.

## Additional information

**Competing financial interests:** The authors declare no competing financial interests.

