## [Peer Review File · Nature Communications]

Reviewers' comments:

Reviewer #1 (Remarks to the Author):

This is a nice paper. De novo functional predictions of proteins is an extremely important and challenging aim. Here the authors not only present a computational method but test some of its predictions by experiment with excellent results. Overall, the results are impressive and deserve publication in Nature Communication. The paper is also well written and illustrated.

I have two minor comments.

First, from what I understand the authors focus on nucleic-acid binding proteins. From what I know, electrostatics and H-bonds can play very important roles in their interactions even though hydrophobic interactions can also contribute to significant extents. The ratios of these contributions in protein-protein interactions differ and depend on the interaction type (e.g. homodimers vs transient interactions). Since here the authors highlight the method as a general one, it would be nice to discuss this point and if possible to also present some results for non NA-binding proteins.

Second, the authors discuss the importance of their biophysical properties. It would be good to mention which they are in the main text as well as the relative weights that they use and the number of parameters as well. Perhaps it is mentioned, but I could not find it. This would lead to a better grasp of some of the reasons behind the methodological success.

Reviewer #3 (Remarks to the Author):

A. Summary of the key results

Peled et al. present a computational approach to predicting the molecular functions of unannotated proteins. They make an excellent case for the importance of this task (e.g., that 99.6% of known proteins have never been observed/studied in the laboratory) and for using a de novo approach, rather than a homology-based approach, which is the current standard. The key results are: 1) promising results from 3-fold cross-validation experiments assessing the performance of a sequence-based machine learning classifier (Dr. PIP) in predicting whether or not a given protein is likely to bind DNA or RNA; 2) apparently accurate computational predictions (DNA-binding or not) for 19 human proteins that had not previously been annotated as "DNA-binding"; 3) preliminary experimental validation of DNA binding activity (using BH1 assays) for these 19 proteins. A major weakness is that these results do not actually constitute de novo molecular function prediction; the results are for predicting the DNA/RNA binding status of a protein only; predictions for other molecular functions are not attempted.

B. Originality and interest: if not novel, please give references

The study focuses on a challenging problem: functional annotation of proteins. A widely-applicable de novo approach to this problem would be both original and of broad interest. The authors claim that their de novo approach for predicting the molecular functions of proteins is novel because most other published methods for protein function prediction are homology-based. This is only partly true: this paper presents binary classifiers for predicting whether a given protein is a nucleic acid binding protein (yes or no) using a sequence-based machine learning approach that does not rely on sequence homology. Unfortunately, the title, in particular, but also sentences in the abstract, introduction and discussion imply that the proposed de novo approach is generally applicable to predicting a wide range of protein functions. This might be the case, but no support for this claim is presented in the current study, which focuses exclusively on nucleic acid binding.

The authors provide references for several reviews and previously published studies that focus on predicting nucleic acid binding (either DNA or RNA binding) (e.g., 13,16-21). Although some of these methods do rely partly or exclusively on sequence homology to make predictions, most of them, in fact, do not. What is true is that most published and widely used approaches for general functional annotation of proteins are homology-based (and often, inaccurate, as the authors note). But, as stated earlier - this is actually NOT the case for published nucleic acid binding predictors, which are more often based on sequence features, rather than sequence-homology.

C. Data & methodology: validity of approach, quality of data, quality of presentation

Unfortunately, the authors do not provide sufficient details to reproduce or even rigorously evaluate the experiments. The approach is potentially very interesting, but the authors do not provide the URL for their Dr. PIP server. A webserver is available (http://ofranservices.biu.ac.il/sandbox/services/dr_pip/index.htm), but all of the documentation available when I accessed it was actually for a different server (CDRAnalyzer) - not for Dr. PIP. Even discounting this omission, I could not rigorously evaluate the quality of the data presented because the authors do not provide enough information about the following: the exact composition of the datasets (all datasets should have been provided as supplemental data); experimental details regarding how the classifiers were constructed (why were RFs chosen? Optimization?); classifier performance on previously published benchmark datasets; details of the B1H DNA-binding validation experiments (positive and negative controls? reproducibility?); controls for the nuclear localization experiment in Figure 7 (which does not prove DNA binding, in any case). A much more detailed description of the methods and more complete Figure legends (instead of titles only) would have been very helpful.

D Appropriate use of statistics and treatment of uncertainties

& E. Conclusions: robustness, validity, reliability

The performance statistics provided are insufficient in several ways. Although precision-recall curves are useful, it is very difficult to quantitatively compare performance of methods using only these. For example, the AUC of ROC would have been better for comparison. Also, to be useful for the most likely users of Dr. PIP (molecular biologists seeking high specificity or high sensitivity predictions), a supplemental table providing values for Specificity, Sensitivity, MCC, etc, based on a specific threshold, and for a dataset of proteins, would have been more useful. More importantly, PR curves are provided only for comparing the new DNA and RNA classifiers proposed by this group, not for the comparisons of Dr. PIP with methods published by other groups. These comparisons (Figs 1,2,3,4) provide no support for the claims of this paper. The most important experiments - comparisons with other methods to determine whether or not Dr. PIP provides any improvement- are not adequate and are poorly documented. Figure 8 shows the "yes or no" results obtained for 19 human proteins. While the authors are to be commended for the considerable amount of work involved in generating and evaluating the constructs required to perform BH1 assays for assessing the potential DNA binding activity for the proteins in this dataset, it is impossible to conclude anything about the relative performance of the various methods without robust performance evaluation statistics. Still, the authors should be commended because they do not claim that the apparent differences in performance among methods (illustrated in Fig 8) are statistically significant.

F. Suggested improvements: experiments, data for possible revision

To live up to its title, the manuscript would need to demonstrate that the de novo approach presented here is generally applicable, which would require training, testing and rigorous evaluation of a rather large number of binary classifiers to cover a wide range of molecular function of interest. While this study could be considered a "proof-of-principle" with its focus on nucleic acid binding (which is comparatively easy to validate experimentally), the performance comparisons with other methods were performed on a small independent test set, which makes it

very difficult to fairly evaluate the claim that this approach actually performs better than existing approaches, even for DNA binding predictions.

With the inclusion of additional experimental details needed to reproduce this work, additional experiments to more rigorously evaluate the comparative performance of this method for predicting DNA binding relative to other published methods, and with a more suitable title, this study could be of considerable interest to scientists in the field, and Dr. PIP could be valuable to a broad spectrum of biological scientists.

G. References: appropriate credit to previous work?

References to recent advances in protein functional annotation (e.g., Radivojac, P., et al. *Nature Methods* 10(3), 221-227 (2013), also, for several recent examples, see a 2016 special issue of *Methods* <http://www.sciencedirect.com/science/journal/10462023/93/supp/C>) are almost completely absent - and references comparing the "state-of-the-art" in de novo protein structure prediction vs protein function prediction are out of date.

The references cited regarding prediction of nucleic acid binding are generally appropriate, but they are provided mainly for recent studies on predicting DNA binding only. One puzzling omission is discussion the recent availability of genome/proteome-wide data regarding DNA and RNA binding proteins from HTP ChIP and RIP-Seq experiments.

H. Clarity and context: lucidity of abstract/summary, appropriateness of abstract, introduction and conclusions

The manuscript is generally well written, although it contains a surprising number of small typographical errors in the text and even in figure labels - and is missing many essential details. The study is very well motivated in the abstract and introduction, but the methods and results are poorly documented. Most figures have titles, but no complete legends; datasets are described, but are not provided as supplemental data. The title, abstract, introduction and conclusions "over-sell" the potential generality of the method.

We thank the reviewers for the positive overviews of our study and for their criticism. They identified a few weak points in the original manuscript and suggested some clever ways to address them. This allowed us to make this ms stronger. Based on their comments, we performed a series of new analyses, repeated some of the experiments and revised the text considerably.

Below is a point-by-point response to their comments:

Reviewer #1

This is a nice paper. De novo functional predictions of proteins is an extremely important and challenging aim. Here the authors not only present a computational method but test some of its predictions by experiment with excellent results. Overall, the results are impressive and deserve publication in Nature Communication. The paper is also well written and illustrated.

Thanks.

I have two minor comments.

First, from what I understand the authors focus on nucleic-acid binding proteins. From what I know, electrostatics and H-bonds can play very important roles in their interactions even though hydrophobic interactions can also contribute to significant extents. The ratios of these contributions in protein-protein interactions differ and depend on the interaction type (e.g. homodimers vs transient interactions). Since here the authors highlight the method as a general one, it would be nice to discuss this point and if possible to also present some results for non NA-binding proteins.

Thanks for this idea. We added an analysis of the composition of the protein-NA interfaces. Positively charged residues and residues that can form H-bonds are overrepresented in these interfaces. Importantly, the composition of predicted interface residues is virtually identical to that of observed interfaces. We added the following paragraph and table to the text:

Looking at the amino acid composition of residues in the interface can also provide some insight into the biophysical characteristics that allow NA recognition. Supplementary Table 8 shows the fraction of positive, negative, polar and hydrophobic residues, as well as that of residues that can form hydrogen bonds. It shows these fractions in proteins in general, in the interfaces and in residues predicted by Dr. PIP to bind DNA or RNA. The composition of the predicted interfaces is virtually indistinguishable from that of observed interfaces, indicating that the machine learned the correct composition of interfaces. Not surprisingly, positive residues are strongly overrepresented while negative residues are under-represented, indicating that the most pronounced feature of the interfaces is the electrostatic interaction.

Amino acid type	Positive	Negative	Polar	Hydrophobic	H-bonds
DNA binding proteins	0.157	0.128	0.192	0.406	0.524
Protein-DNA interfaces	0.296	0.074	0.252	0.27	0.684
Prediction	0.294	0.078	0.254	0.24	0.675
RNA binding proteins	0.162	0.125	0.181	0.409	0.512

Protein RNA interfaces	0.31	0.08	0.205	0.287	0.647
Prediction	0.311	0.085	0.205	0.265	0.641

Supplementary Table 8. Types of amino acids in protein-NA interfaces and in our predicted binding sites, compared to their propensities in the protein in general.

Second, the authors discuss the importance of their biophysical properties. It would be good to mention which they are in the main text as well as the relative weights that they use and the number of parameters as well. Perhaps it is mentioned, but I could not find it. This would lead to a better grasp of some of the reasons behind the methodological success.

In the revised version we put greater emphasis on the list of features and provide more details about it. We included these sentences in the main text:

To train these machines, each residue was described by 246 features that included: predicted structural features (secondary structure, solvent accessibility and disorder) as well as sequence features (type of amino acid, sequence neighbors and their characteristics, conservation, entropy. See Methods for details).

And:

To train these machines, each protein was described by 15 features representing the distribution and the scores of the prediction of the per-residue machines (e.g. how many of the residues got a certain score. See Methods).

The relative contribution of each feature is calculated by the RF algorithm. We also added the following text (4.1.2 in Methods).

Each residue was described as a set of calculated and predicted numeric attributes. They include: Evolutionary conservation extracted per each residue and the number of proteins in the alignment, extracted from HSSP³⁴; predicted solvent accessibility and secondary structure from PROF-phd³⁵; predicted disorder from MD³⁶. In addition, since a residue is affected by its neighbors, sliding windows were used as descriptors of each residue. The properties of the surrounding residues were also included in the vector. We explored different sizes of sliding windows for each attribute to determine the optimal window. For the secondary structure and disorder, a window size of seven residues, centered by the residue for which we wish to provide prediction, gave the best performance. For the evolutionary profile, a window size of nine residues was used. Each vector contained a total of 246 features. Each such vector described one residue and was labeled: '1' or '0' for a binding or a non-binding residue, respectively.

Reviewer #3 (Remarks to the Author):

A. Summary of the key results □

Peled et al. present a computational approach to predicting the molecular functions of unannotated proteins. They make an excellent case for the importance of this task (e.g., that 99.6% of known proteins have never been observed/studied in the laboratory) and for using a de novo approach, rather than a homology-based approach, which is the current standard. The key results are: 1) promising results from 3-fold cross-validation experiments assessing the performance of a

sequence-based machine learning classifier (Dr. PIP) in predicting whether or not a given protein is likely to bind DNA or RNA; 2) apparently accurate computational predictions (DNA-binding or not) for 19 human proteins that had not previously been annotated as "DNA-binding"; 3) preliminary experimental validation of DNA binding activity (using BH1 assays) for these 19 proteins. A major weakness is that these results do not actually constitute *de novo* molecular function prediction; the results are for predicting the DNA/RNA binding status of a protein only; predictions for other molecular functions are not attempted.

We address this comment in details below.

B. Originality and interest: if not novel, please give references

The study focuses on a challenging problem: functional annotation of proteins. A widely-applicable *de novo* approach to this problem would be both original and of broad interest. The authors claim that their *de novo* approach for predicting the molecular functions of proteins is novel because most other published methods for protein function prediction are homology-based. This is only partly true: this paper presents binary classifiers for predicting whether a given protein is a nucleic acid binding protein (yes or no) using a sequence-based machine learning approach that does not rely on sequence homology. Unfortunately, the title, in particular, but also sentences in the abstract, introduction and discussion imply that the proposed *de novo* approach is generally applicable to predicting a wide range of protein functions. This might be the case, but no support for this claim is presented in the current study, which focuses exclusively on nucleic acid binding.

Thanks for this comment. We obviously use DNA and RNA binding as test cases, to demonstrate how biophysical features can predict molecular function *de novo*. We changed the title to: ***De-novo* protein function prediction - DNA binding and RNA binding as test cases**

We also modified the text in several places to clarify this. As for the importance of NA-binding prediction: Obviously, the end goal of the field is to be able to provide one day a comprehensive and detailed annotation of every and all aspects of the function of each protein. However, one has to consider the state of the art: GO defines DNA-binding and RNA-binding as molecular functions that have many daughters in the GO tree. Currently, determining whether a protein has a function that involves NA binding remains a major challenge. Novel **experimental** approaches for addressing this question are published in the top journals (e.g. Hellman and Fried (2007) Nature protocols, 2(8), 1849-1861, Lehlanc and Moss (1994) DNA-protein interactions: principles and protocols, 1-10, Zhang, C., & Darnell, R. B. (2011) Nature biotechnology, 29(7), 607-614). We show that we can do well in this task with computational analysis. We also show that Dr. PIP can succeed with proteins that were never studied before. Such proteins are a challenge even for experimental methods, as most of them require a specific antibody against the protein or successful recombinant expression and purification of the protein. Dr. PIP requires only the sequence. Thus, Dr. PIP constitutes a major step forward in function prediction.

The authors provide references for several reviews and previously published studies that focus on predicting nucleic acid binding (either DNA or RNA binding) (e.g., 13,16-21). Although some of these methods do rely partly or exclusively on sequence homology to make predictions, most of them, in fact, do not. What is true is that most published and widely used approaches for general functional annotation of proteins are homology-based (and often, inaccurate, as the authors note). But, as

stated earlier - this is actually NOT the case for published nucleic acid binding predictors, which are more often based on sequence features, rather than sequence-homology.

There are two types of prediction methods that we discuss: (i) methods that predict NA binding sites, and (ii) methods for identifying NA binding proteins. The reviewer is right to point out that some of the methods for predicting NA binding sites rely on sequence and structural features. However, these methods do not predict whether a protein binds NA or not. Rather, they are designed to analyze proteins that are already known to bind NA, and identify the residues that bind the DNA or RNA. But this is not the case for methods of the second kind, namely those that analyze an un-annotated sequence to predict whether it binds DNA. Most of them rely, either implicitly or explicitly, on homology or on structural similarity to annotated proteins. This becomes clear in the revised version, as we show that existing methods do not perform well on novel NABPs. By “novel” we mean proteins that cannot be reliably identified based on homology to known NABPs. We built several different sets of proteins to show that, beyond the 19 ORFans we used in the original version (for which we show that Dr. PIP separated correctly the DNA binders from the non-binders and that no other methods succeeded in that). In the revised version we add a second set of hundreds of DBPs that are not members of any superfamily that is known to bind DNA and have no sequence motif or profile that is associated with DNA binding. We show that existing methods perform rather poorly on this set, while Dr. PIP’s performance is almost identical to its performance on other sets.

Unfortunately, the authors do not provide sufficient details to reproduce or even rigorously evaluate the experiments.

We extended the methods section to include every detail of the development of the method, from the construction of the training set to testing and validation. We included all the sets of proteins we used as supplementary materials.

The approach is potentially very interesting, but the authors do not provide the URL for their Dr. PIP server. A webserver is available (http://ofranservices.biu.ac.il/sandbox/services/dr_pip/index.html), but all of the documentation available when I accessed it was actually for a different server (CDRAnalyzer) - not for Dr. PIP.

We didn’t include this URL in the manuscript or in our homepage, as it was under construction. We are sorry for not blocking it for external access. Our goal was to launch a server later, after this manuscript is accepted. However, given the comment of the reviewer, we accelerated the development of the server and it is now available at http://ofranservices.biu.ac.il/site/services/dr_pip. Note that we labeled the server as beta. This will be removed if and when the ms is accepted. The server is also available at www.ofranlab.org under services.

Even discounting this omission, I could not rigorously evaluate the quality of the data presented because the authors do not provide enough information about the following: the exact composition of the datasets (all datasets should have been provided as supplemental data);

We now provide this information in supplementary tables 5 and 6.

experimental details regarding how the classifiers were constructed (why were RFs chosen? Optimization?);

We now address this in the methods section. We tested several ML algorithms (ANN, SVM). RF performed better. We optimized the number of trees and the features by parameter sweeping.

classifier performance on previously published benchmark datasets;

We added Table 3 that shows the performance of Dr. PIP on a recent benchmark set and compare it to other methods. Dr. PIP outperforms the other methods (as expected, given its ability to predict function *de novo*).

details of the B1H DNA-binding validation experiments (positive and negative controls? reproducibility?); controls for the nuclear localization experiment in Figure 7 (which does not prove DNA binding, in any case). A much more detailed description of the methods and more complete Figure legends (instead of titles only) would have been very helpful.

We repeated the experiment on FGF14 subcellular localization in human cell, generated better figures, and added more details about the controls. We also added a sentence that clarifies that the experiment does not show DNA binding (which was shown by the B1H) but rather demonstrates the biological *in-vivo* feasibility of DNA binding by showing that FGF14 is localized to the nucleolus.

The performance statistics provided are insufficient in several ways. Although precision-recall curves are useful, it is very difficult to quantitatively compare performance of methods using only these. For example, the AUC of ROC would have better for comparison. Also, to be useful for the most likely users of Dr. PIP (molecular biologists seeking high specificity or high sensitivity predictions), a supplemental table providing values for Specificity, Sensitivity, MCC, etc, based on a specific threshold, and for a dataset of proteins, would have been more useful.

We (and others in the function prediction field) challenge that PRC is the most relevant measure for performance in cases like this, given the imbalance of real-world datasets (which can contain >90% negative samples). However, we acknowledge that other measures may be of interest. To address the reviewer's concern we added supplementary tables, both for DNA and one for RNA, with AUC, sensitivity, specificity, FPR, accuracy and MCC. The tables show these values for multiple cutoffs. We also added two ROC Figures for DBPs and RBPs predictions as well as a table and PRCs of comparisons to other methods (See response to next point as well).

More importantly, PR curves are provided only for comparing the new DNA and RNA classifiers proposed by this group, not for the comparisons of Dr. PIP with methods published by other groups. These comparisons (Figs 1,2,3,4) provide no support for the claims of this paper. The most important experiments - comparisons with other methods to determine whether or not Dr. PIP provides any improvement- are not adequate and are poorly documented. Figure 8 shows the "yes or no" results obtained for 19 human proteins. While the authors are to be commended for the considerable amount of work involved in generating and evaluating the constructs required to perform BH1 assays for assessing the potential DNA binding activity for the proteins in this dataset, it is impossible conclude anything about the relative performance of the various methods without

robust performance evaluation statistics. Still, the authors should be commended because they do not claim that the apparent differences in performance among methods (illustrated in Fig 8) are statistically significant.

In the revised version we elaborate on the advantages of Dr. PIP over existing methods. To do this we use several different datasets and an array of measures.

1. We use a previously published benchmark dataset that compared the best performing methods. We added Dr. PIP to this benchmark and show that in all the measurements Dr. PIP outperforms other methods (Table 3).
2. When possible, we used our dataset to produce a PRC comparing Dr. PIP to the methods that allow large scale submissions. We show that Dr. PIP outperforms the others across the ranges of precision and recall.
3. We use a dataset of novel DBPs to present a PRC just for these difficult proteins. We show that other methods perform poorly, while Dr. PIP's performance on this key dataset is similar to its performance of the validation set. This, we believe, provides not only the statistical validity that is needed, but also an explanation to the superior performance of Dr. PIP. It does better than others because of its performance on novel proteins. It does better on novel proteins, because it relies on biophysical features, not on sequence (or structural) similarity.

F. Suggested improvements: experiments, data for possible revision □

To live up to its title, the manuscript would need to demonstrate that the de novo approach presented here is generally applicable, which would require training, testing and rigorous evaluation of a rather large number of binary classifiers to cover a wide range of molecular function of interest. While this study could be considered a "proof-of-principle" with its focus on nucleic acid binding (which is comparatively easy to validate experimentally),

We now make clear that this study is, as the reviewer puts it, a proof-of-principle.

the performance comparisons with other methods were performed on a small independent test set, which makes it very difficult to fairly evaluate the claim that this approach actually performs better than existing approaches, even for DNA binding predictions.

We now compare the performance on a large benchmark dataset that was published by a third party. We also add analysis on two other large sets, including a set of novel DBPs.

With the inclusion of additional experimental details needed to reproduce this work,

Done.

additional experiments to more rigorously evaluate the comparative performance of this method for predicting DNA binding relative to other published methods,

Done. See new figures, tables and supplementary materials.

and with a more suitable title,

Done.

this study could be of considerable interest to scientists in the field, and Dr. PIP could be valuable to a broad spectrum of biological scientists.

G. References: appropriate credit to previous work?

References to recent advances in protein functional annotation (e.g., Radivojac, P., et al. Nature Methods 10(3), 221-227 (2013), also, for several recent examples, see a 2016 special issue of Methods <http://www.sciencedirect.com/science/journal/10462023/93/supp/C>) are almost completely absent - and references comparing the "state-of-the-art" in de novo protein structure prediction vs protein function prediction are out of date.

Thanks for pointing this oversight. This important paper was mentioned in our cover letter but we somehow failed to include it in the final version of the ms. We added it now. We also added additional references the reviewer mentioned as well as other, more up-to-date references we collected.

The references cited regarding prediction of nucleic acid binding are generally appropriate, but they are provided mainly for recent studies on predicting DNA binding only. One puzzling omission is discussion the recent availability of genome/proteome-wide data regarding DNA and RNA binding proteins from HTP ChIP and RIP-Seq experiments.

A reference to a recent RBPs prediction was added (Yan, J. et al. (2015) Brief Bioinform).

We also added the following references: Si, J. et al. (2015) International journal of molecular sciences, 16(11), 26303-26317, Robertson, G. et al. (2007) Nature methods, 4(8), 651-657, Zhao, J. et al. (2010) Molecular cell, 40(6), 939-953, Park, P. J. (2009) Nature Reviews Genetics, 10(10), 669-680, Gilfillan, G. D. et al. (2012) BMC genomics, 13(1).

H. Clarity and context: lucidity of abstract/summary, appropriateness of abstract, introduction and conclusions. The manuscript is generally well written, although it contains a surprising number of small typographical errors in the text and even in figure labels - and is missing many essential details.

We proofread the manuscript carefully.

The study is very well motivated in the abstract and introduction, but the methods and results are poorly documented.

We added substantial amount of details.

Most figures have titles, but no complete legends; datasets are described, but are not provided as supplemental data. The title, abstract, introduction and conclusions "over-sell" the potential generality of the method.

Thanks. We corrected those as described above.

We hope that as we addressed these concerns, you will now find that this improved manuscript is of "considerable interest to scientists in the field" and "valuable to a broad spectrum of biological scientists."

REVIEWERS' COMMENTS:

Reviewer #1 (Remarks to the Author):

I did not test the server, nor go into statistical detail and comparisons with other approaches. But going over the the responses of the authors they seem OK and I have no further comments.

Reviewer #3 (Remarks to the Author):

The authors have performed additional experiments, made their Dr. PIP server available and extensively revised the manuscript to address all of my earlier concerns. Bravo - very nice work!